# Quality of life after resection of a meningioma—A cross-cultural comparison of Indian and Australian patients

Verena Schadewaldt[1,2]*, Sandhya Cherkil[3], Dilip Panikar[3], Katharine J. Drummond[1,2]

**1** Department of Surgery, Faculty of Medicine, Dentistry and Health Sciences, University of Melbourne, Melbourne, Australia, **2** Department of Neurosurgery, The Royal Melbourne Hospital, Melbourne, Australia, **3** Department of Neuroscience, Aster Medcity, Kerala, India

* verena.schadewaldt@unimelb.edu.au

**Data Availability Statement:** All files are available from figshare data repository under the URL: https://doi.org/10.26188/624ecdc85d240.

## Abstract

### Purpose

To compare health-related quality of life (HRQoL) and symptom burden following meningioma resection in patients from two samples from Australia and India. This will add to the body of data on the longer-term consequences of living with a meningioma in two socio-economically and culturally different countries.

### Methods

The European Organisation for Research and Treatment of Cancer (EORTC) Core Quality of Life Questionnaire (QLQ-C30), Brain Neoplasm Module (QLQ-BN20) and the Hospital Anxiety and Depression Scale (HADS) were administered to 159 Australian and 92 Indian meningioma patients over 24 months postoperative. A linear mixed model analysis identified differences between groups over time.

### Results

Australian patients reported better physical functioning in the early months after surgery (T1: mean diff: 19.8, p<0.001; T2: mean diff: 12.5, p = 0.016) whereas Indian patients reported better global HRQoL (mean: -20.3, p<0.001) and emotional functioning (mean diff:-15.6, p = 0.020) at 12–24 months. In general, Australian patients reported more sleep and fatigue symptoms while Indian patients reported more gastro-intestinal symptoms over the 2-year follow-up. Future uncertainty and symptoms common for brain tumour patients were consistently more commonly reported by patients in Australia than in India. No differences for depression and anxiety were identified.

### Conclusion

This is the first cross cultural study to directly compare postoperative HRQoL in meningioma patients. Some differences in HRQoL domains and symptom burden may be explained by culturally intrinsic reporting of symptoms, as well as higher care support from family

**Funding:** KD was supported by a Melbourne Medical School Strategic Grant for Outstanding Women (University of Melbourne). https://www.unimelb.edu.au/ The funder had no role in study design, data collection and analysis, decision to publish, or preparation of the manuscript.

**Competing interests:** The authors have declared that no competing interests exist.

members in India. Although there were differences in some HRQoL domains, clinically meaningful differences between the two samples were less common than perhaps expected. This may be due to an Indian sample with high literacy and financial resources to afford surgery and follow up care.

## Introduction

Meningiomas are the most common brain tumours in adults. They account for 37% of all brain tumours [1] and ten-year survival rates range from 77%-81% for grade 1, 63%-71% for grade 2 and 15%-23% for grade 3 tumours [2, 3]. As most meningiomas are benign, survival rates are extended compared to other brain tumors [4]. Additionally, advances in surgical techniques and adjuvant therapy have led to dramatic improvements in care over the past two decades. Consequently, with extended survival and reduction of gross neurological morbidity, measures of treatment success have appropriately shifted to more patient-centred metrics, including health-related quality of life (HRQoL). HRQoL is a self-assessed, multidimensional concept that encompasses the physical, emotional, role, social and cognitive components of quality of life (QoL) associated with illness and its treatment.

A few studies have compared HRQoL of meningioma patients to a normative population with scores expectedly better in the normative group [5–7] A systematic review on meningioma HRQoL that included 19 studies from 12 countries highlighted that, in particular, physical, role, emotional, cognitive and social functioning were lower in the meningioma group compared to a normative population [8], however results for global HRQoL and fatigue were not consistent. The review concluded that a comparison of outcomes was difficult due to the use of different instruments to measure HRQoL [8]. We report the first cross cultural comparison of HRQoL outcomes in a sample of meningioma patients in Australia and India, using the European Organisation for Research and Treatment of Cancer (EORTC) QLQ-C30 and BN20, as well as the Hospital Anxiety and Depression Scale (HADS).

India and Australia have inherently different health care and social systems. In 2018, health care expenditure in Australia was 9.25% of gross domestic product (GDP) compared to 3.66% in India [9]. Patient out-of-pocket expenses vary greatly, with Indian patients paying out-of-pocket for most surgeries and Australian patients usually covered by private or government health insurance. The index for impoverishing expenditure shows that 0.1% of Australian and 26.7% of Indian inhabitants would be at risk of extreme poverty when requiring surgical care such as meningioma resection [10]. Australian patients have better access to health professionals with more doctors and nurses available per inhabitant, however there is great variation across Indian states and regions [11, 12]. While these statistics suggest disadvantage for some people in India to access and receive health care, it is unclear whether and how this is reflected in HRQoL and symptom burden in patients undergoing meningioma surgery.

The social systems in the two countries also differ. Although in transformation, the majority of people in India practice a collectivistic lifestyle, intrinsically based on family support and interdependence [13]. This system understands the family as a vast resource for the sick or disabled [14]. People in Australia conform to a more individualistic independent system, where, in many cases, recovery from illness takes place in institutional care settings [15] or at home with limited support from others. It is unknown how these differences may affect HRQoL and perceptions of symptom burden.

We aim to compare HRQoL, symptom burden and anxiety and depression in postoperative meningioma patients from Australia and India, using the results of two independently collected datasets that, however, used the same HRQoL instruments.

## Materials and methods

This paper presents the opportunistic secondary analysis of data that were collected for two separate studies in Australia and India that used the same HRQoL instruments in meningioma patient cohorts. Both studies used convenience samples and timing for follow up assessments differed but were matched for the analysis.

### Data collection—Australia

The data for this study were collected by medical students in an assigned research rotation under direct supervision of KD as part of a prospective longitudinal observational study, including patients ≥ 18 years who had been admitted for intracranial meningioma resection and attended routine follow-up at the Neuro-Oncology and Neurosurgery Outpatient clinics and Private clinics at the Royal Melbourne Hospital. Patients with other brain or spine lesions or patients with neurofibromatosis type I or II were excluded. Patients needed to complete the survey in English without the help of a proxy. The sample included patients from two large tertiary institutions that service remote, rural, and urban populations and both public (government insurance) and private (insured) patients.

Eligible patients were approached opportunistically by research assistants while attending the clinic for an appointment between February 2014 to March 2020. No data was recorded on the number of patients who were approached and declined to participate.

After giving written informed consent, patients completed the questionnaire while at the hospital for their appointment. In some cases, patients completed the questionnaire at home and returned it by mail. Patients could enter the study at any point postoperative and completed the questionnaire at every subsequent visit to the hospital for longitudinal monitoring. Postoperative visits occurred at 6 weeks, 3 months and then 6–12 month intervals until discharge, usually at 7 years postoperative if no evidence of tumour on MRI, or indefinitely if residual tumour and until radiological progression requiring further treatment, advanced age or death. No data were collected preoperatively.

Demographic details and tumour characteristics were also collected from the medical record and the prospectively maintained Royal Melbourne Hospital Brain Tumour Database (part of the Australian Comprehensive Cancer Outcomes and Research Database [ACCORD]). This study was approved by the Melbourne Health Human Research Ethics Committee in 2013 (no. 2013.246).

### Data collection—India

Meningioma patients diagnosed between October 2014 and December 2019 at Aster Medcity Hospital, Kochi, Kerala, underwent detailed neuropsychological assessment, including HRQoL, at their preoperative workup which is part of the standard institutional treatment protocol. The sample included consenting, literate, English-speaking patients over 18 years of age. Patients who exhibited frank psychiatric disturbances, severe sensory/motor impairments and comprehension difficulties or who did not agree to take the assessment were excluded. Eligible patients were directed to the neuropsychologist after the treatment protocol was explained to them by the neurosurgeon. Quality of life, depression and anxiety questionnaires were administered at postoperative follow up at 1, 3, 6, 12 and up to 24 months. Patient demographics and tumour details were recorded during the clinical interaction as well as during the administration of the questionnaires by SC. These assessments were undertaken in Malayalam, the local language, without a proxy, despite the patient's ability to speak English. The study was approved by the Institutional Ethics Committee.

## Instruments

**EORTC QLQ-C30.** HRQoL was assessed using the EORTC QLQ-C30, Version 3.0 [16]. This validated questionnaire comprises 30 items determining a global HRQoL scale, five functional scales, three symptom scales and six symptom single-item measures [17]. The global HRQoL scale assesses current health status and overall HRQoL. The functional scales assess physical, role, emotional, cognitive and social functioning. The symptom scales and items assess fatigue, nausea and vomiting, pain, dyspnea, insomnia, appetite loss, constipation, diarrhea and financial difficulties. Rating occurs on a four-point Likert scale (not at all–a little–quite a bit–very much) for all items apart from the two global HRQoL items, which are rated on a seven-point scale (very poor to excellent). Scores are transformed to a score from 0–100 with higher scores indicating better global HRQOL and functioning, but higher symptom burden according to EORTC guidelines [17]. Indian patients completed the QLQ-C30 in the validated Malayalam version [18].

**EORTC BN20.** To more specifically capture HRQoL in patients with brain cancer the EORTC QLQ-BN20 (brain cancer module) was used [19]. The BN20 comprises 20 items aggregated into four scales and seven single-item measures. The scales assess future uncertainty, visual disorder, motor dysfunction, and communication deficit. The single items assess headaches, seizures, drowsiness, hair loss, itchy skin, weakness of legs and bladder control. Items are rated on a four-point Likert scale (not at all–a little–quite a bit–very much) and transformed scores range from 0–100 with higher scores indicating higher symptom burden and therefore lower HRQoL. Indian patients completed the BN20 in a locally translated Malayalam version.

**HADS.** Anxiety and depression were assessed using the HADS, a validated tool to screen for anxiety and depression in the hospital or community health setting [20]. It contains 14 items with seven items for the depression subscale and seven for the anxiety subscale. Items are rated by four response options that are scored from 0–3 with some items being in reverse order. The sum of the scores for each subscale is scored separately and results range from normal (0–7) to borderline abnormal (8–10) to abnormal (11–21) [21]. Indian patients completed the HADS in the validated Malayalam version [22]. The HADS was introduced later to the Australian study, so fewer patients completed it, leading to some missing data.

**Socioeconomic status.** Socioeconomic (SES) status for Australian patients was derived from The Index of Relative Socio-economic Advantage and Disadvantage (IRSAD) [23] based on the patient's postcode. For Indian patients, the SES was loosely based on the Modified Kuppuswamy Classification [24]. The 10-point scale of the IRSAD was reduced to a 5-point scale to match the Indian Classification by combining two points to one.

## Data analysis

The two datasets were merged in Excel and analysed with IBM SPSS Statistics, Version 26.0. Time points 1 (1–2 months), 2 (3–6 months), 3 (7–12 months) and 4 (13–24 months) for survey completion postoperative were created to enable comparison of outcomes in the Indian and Australian sample using a balanced number of surveys in each sample at each time point. Previous research in meningioma patients identified that HRQoL effects are stable for years after surgery, which justifies the longer intervals for later time points in this analysis [8, 25]. For cases where a patient completed more than one questionnaire within a time point, the first questionnaire was used.

Frequencies and proportions for categorical variables and means and standard deviations for continuous variables are presented. A linear mixed model (LMM) analysis with variance component structure was applied to identify differences between the two samples. These

models allow for fixed and random effects and can be used in cases where data may not be entirely independent as in our case, and where multiple measurements were undertaken on the same person and correlations may exist between these measurements. We had minimal missing data: Across all time points and survey items missing data were below 0.5% for the EORTC QLQ-C30 and BN20. Of those patients who completed EORTC surveys, there were 13 in the Indian sample that did not complete the HADS due to fatigue or refusal to respond and 110 in the Australian sample who did not receive the HADS, leaving 138 (India) and 184 (Australia) HADS for analysis. There were no missing items on the available HADS surveys in both groups. LMM procedures prevent listwise deletion due to missing data, maximizing the use of available data. The assumption of constant variance was reasonably satisfied when tested by plots of residuals versus predicted values.

Independent factors (outcome variables) were HRQoL, symptom burden, anxiety and depression. Dependent factors (predictors) were time point and country. Potentially confounding factors such as age, gender, WHO grade, tumour location and lateralization, SES, employment status and relationship status were added to the model if they showed a significant association with the outcome variables. In our model, fixed effects were time point, country, confounding factors if significant and interactions between time points and country. Statistical significance was assumed at p-value ≤ 0.05.

In addition, minimum clinically important differences or "clinically meaningful difference" (CMD) between groups were explored for the Australian and Indian samples. Using normative reference data for a standardised questionnaire, one can derive CMD, typically taken as a proportion of the standard deviation (SD) of the baseline HRQOL score for the population. This is an indication of the degree of change in HRQoL that should be deemed 'significant' to the patient and therefore warrant clinical attention regardless of statistical significance. Based on previous publications, changes of +/- 10 units were chosen as CMD for most scales of the QLQ-C30 [26, 27]. There are no established CMD scores for the BN20 [27]. A change of +/- 1.6 units indicates a CMD for either scale of the HADS [28].

### Inclusivity in global research

Additional information regarding the ethical, cultural, and scientific considerations specific to inclusivity in global research is included in the S1 Checklist.

### Results

There were 159 and 92 patients in the Australian and Indian sample respectively. Patient and tumour characteristics of the two samples were generally comparable with expected female predominance (Table 1). Age was significantly different, likely reflecting the older Australian general population, with a median age of 38.7 years, compared to 28.1 years for India [29]. The significant difference for WHO grades likely reflects difference in local histopathological practice although a true difference between the two populations cannot be excluded, for instance due to genetic variations or differences in presentation for treatment. US data show the proportion of WHO grade 1 meningiomas to be at 80.6%, grade 2 at 17.4% and grade 3 at 2.1%, which lies between the values of our two samples [30]. Employment, relationship status and SES showed significant differences between the groups with Australian patients having a fifth of patients in the three top SES categories but also 14% coming from the lowest SES, whereas no Indian patients were from the lowest SES category. More than half of the Indian sample was unemployed, mostly due to female dominance in the study, whereas Australian patients comprised 35% of retirees. Indian patients were less likely to be divorced or separated.

**Table 1. Patient and tumour characteristics.**

|  | Australia n = 159 | | India n = 92 | | |
|---|---|---|---|---|---|
|  | **Mean (SD)** | **Min/Max** | **Mean (SD)** | **Min/Max** | **p-value**[a] |
| **Age** | **58.00 (12.8)** | **31/86** | **52.3 (13.0)** | **19/78** | **<0.001** |
|  | **Frequency** | **Percent** | **Frequency** | **Percent** | **p-value**[b] |
| **Gender** |  |  |  |  | 0.084 |
| Male | 36 | 22.6 | 30 | 32.6 |  |
| female | 123 | 77.4 | 62 | 67.4 |  |
| **Tumour location** |  |  |  |  | 0.285 |
| Anterior skull base | 52 | 32.7 | 29 | 31.5 |  |
| Posterior skull base | 34 | 21.4 | 13 | 14.1 |  |
| Convexity | 73 | 45.9 | 50 | 54.3 |  |
| **Tumour lateralisation** |  |  |  |  | 0.711 |
| Left | 72 | 46.2 | 39 | 42.9 |  |
| Right | 67 | 42.9 | 39 | 42.9 |  |
| Midline/bilateral | 17 | 10.9 | 13 | 14.3 |  |
| **WHO grade** |  |  |  |  | <0.001 |
| 1 | 144 | 90.6 | 58 | 63.0 |  |
| 2/3 | 14 | 8.8 | 34 | 37.0 |  |
| **Socio-economic status** |  |  |  |  | <0.001 |
| Highest | 34 | 21.7 | 8 | 8.8 |  |
| Upper mid-level | 44 | 28.0 | 37 | 40.7 |  |
| Mid-level | 42 | 26.8 | 41 | 45.1 |  |
| Lower mid-level | 15 | 9.6 | 5 | 5.5 |  |
| Lowest | 22 | 14.0 | 0 | 0 |  |
| **Relationship** |  |  |  |  | 0.016 |
| Married/de facto | 108 | 67.9 | 74 | 80.4 |  |
| Divorced/separated | 19 | 11.9 | 1 | 1.1 |  |
| Single | 19 | 11.9 | 8 | 8.7 |  |
| Widow/widower | 11 | 6.9 | 9 | 9.8 |  |
| Other relationship | 2 | 1.3 | 0 | 0 |  |
| **Employment** |  |  |  |  | <0.001 |
| Full time | 39 | 25.8 | 28 | 30.4 |  |
| Part time | 21 | 13.9 | 6 | 6.5 |  |
| Casual | 10 | 6.6 | 0 | 0 |  |
| Unemployed | 27 | 17.9 | 51 | 55.4 |  |
| Retired | 54 | 35.8 | 7 | 7.6 |  |

[a] independent samples t-test

[b] Pearson's chi Square

## Comparison of mean scores

**QLQ-C30 –Global HRQoL and functional scales.** Fig 1 shows scores of the Indian and Australian samples in comparison to a normative (European) population [31], which, as expected, demonstrates impaired HRQoL across all domains for our samples. A CMD (improvement or deterioration) in comparison to the European mean is also depicted. While there are within-group differences at certain time points for some variables, the focus of this study is on between-group differences and these are presented here. Global HRQoL remained stably reduced over the study period for Australian patients (between 66.6 and 70.0) and

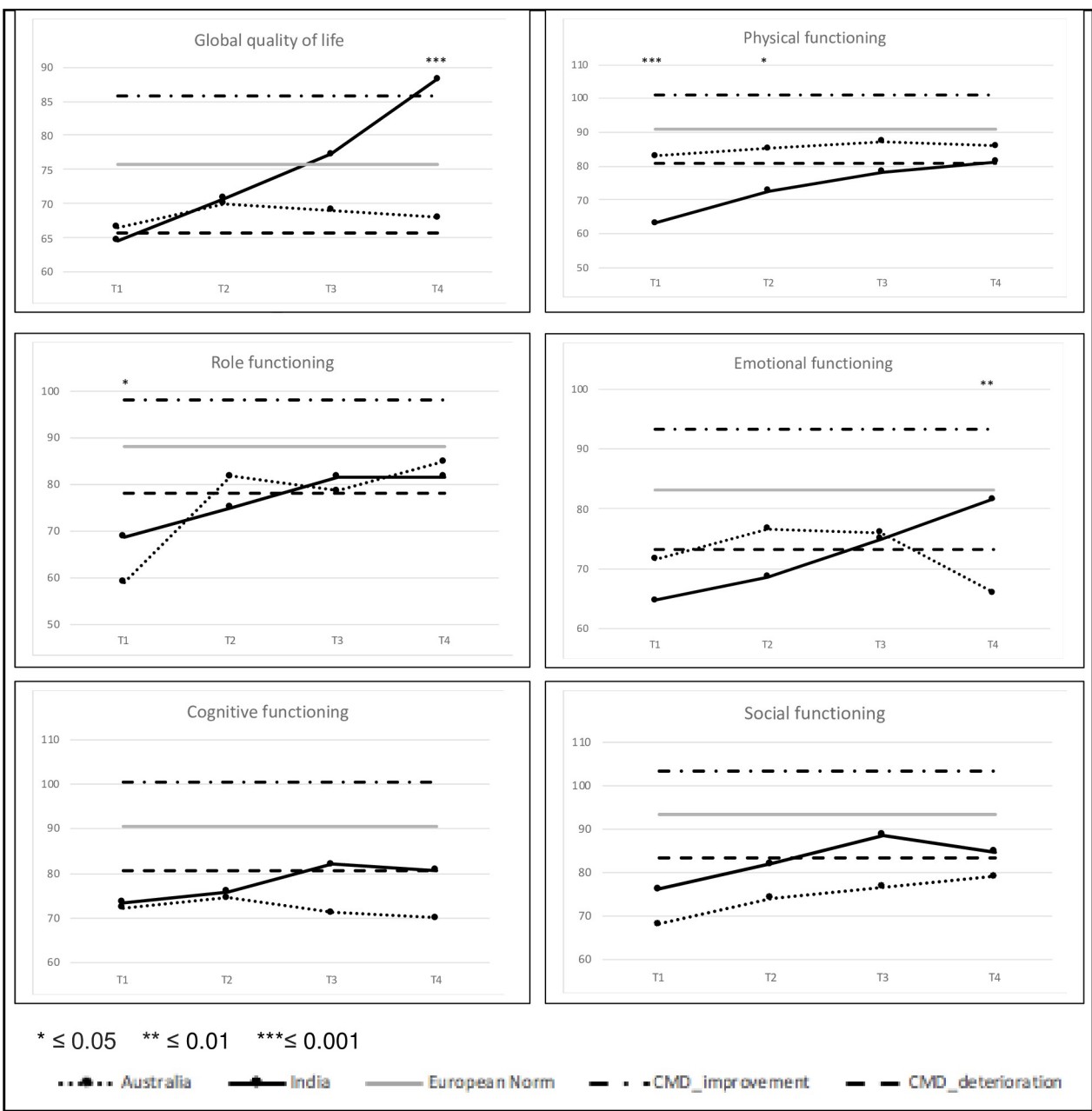

**Fig 1. QLQ-C30 function scales and clinically meaningful difference (CMD) to the European normative population.**

markedly improved over time to better than the normative population for Indian patients (from 64.6 to 88.2) in the first 24 months postoperative, resulting in a significant difference at T4 (mean: -20.3, p<0.001) (data shown in S1 Table). The interaction effect is statistically significant, indicating that the difference in global HRQoL depends on when it was assessed.

Similarly, physical function remained stably (but not clinically meaningfully) reduced over the study period for Australian patients (between 82.9 and 87.3). However, for the Indian sample physical function was initially significantly reduced (T1: mean diff: 19.8, p<0.001; T2: mean diff: 12.5, p = 0.016) but improved over the study period to a similar level to the Australian sample (from 63.1 to 81.2). Differences at T1 and T2 show a CMD with Australian patients

reporting better physical function. Emotional function was initially similar in both countries with a steep improvement reported by Indian patients at T4, which is statistically and clinically significantly different to Australian patients (mean diff: -15.6, p = 0.020). Role function was better in Indian patients at T1 (mean diff: -9.8, p = 0.044) and had similar ratings across groups for later time points. Cognitive and social function showed CMDs towards the later time points with higher functioning in patients in India.

**QLQ C30—Symptom scales.** In general, the frequency of differences in symptom burden reduced over time (Fig 2). The nine symptom scales (pain scale not shown in Fig 2) showed 3,

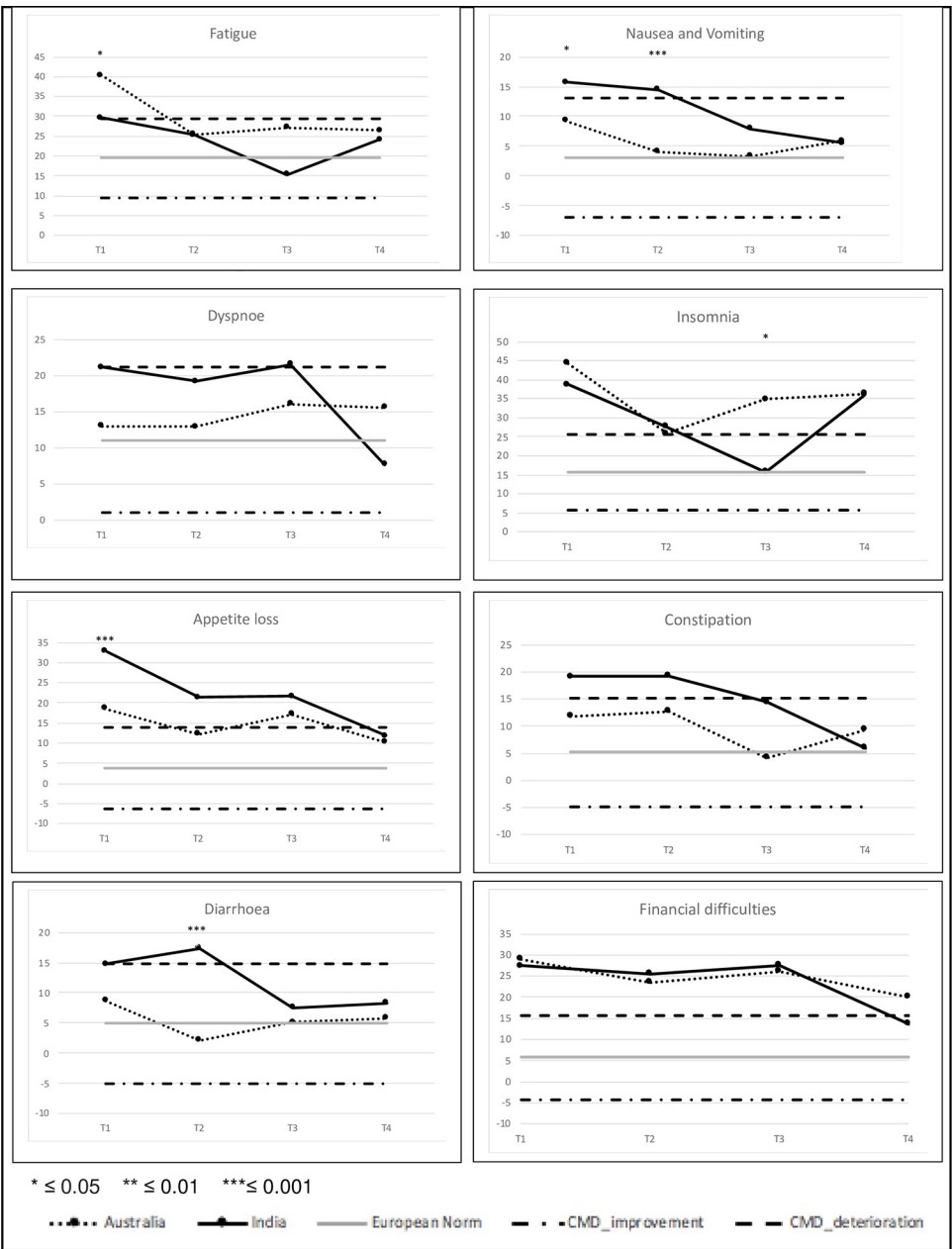

**Fig 2. QLQ-C30 symptom scales and clinically meaningful difference (CMD) to the European normative population.**

2, 1 and 0 significant differences at T1, T2, T3 and T4, respectively. At T1 Australian patients reported more fatigue (mean diff: 10.6, p = 0.017) whereas Indian patients reported higher incidence of nausea/vomiting (mean diff: -6.4, p = 0.040) and appetite loss (mean diff: -14.4, p = 0.002). At T2 symptoms of nausea/vomiting (mean diff: -10.4, p = 0.003) and diarrhea (mean diff: -15.3, p<0.001) were significantly higher in the Indian sample. At T3 insomnia was higher in the Australian sample (mean diff: 19.0, p = 0.038) with fatigue also reaching a CMD at this point (mean diff: 11.8). There were no differences in symptom burden at T4.

According to the mixed model results (which accounts for interaction effects) fatigue, pain, dyspnoea, insomnia, constipation and financial difficulties did not differ overall between the countries (S2 Table). However, constipation (mean diff: -10.2) showed a CMD at T3 with Indian patients reporting higher burden.

**BN20.** BN20 ratings show that patients in Australia report higher symptom burden across all scales and time points, the only exception being seizures at T2 (Fig 3). Many of the symptoms were almost non-existent in the Indian sample. Future uncertainty and headaches were symptoms rated highest in both groups. According to the mixed model results, accounting for interaction effects, all but future uncertainty and seizures differed overall between the countries. Itchy skin and bladder control were statistically significant different across all time points (data shown in S3 Table). Visual disorder, motor dysfunction, communication deficit and headaches were statistically significant at three of four time points. Seizures and weakness of

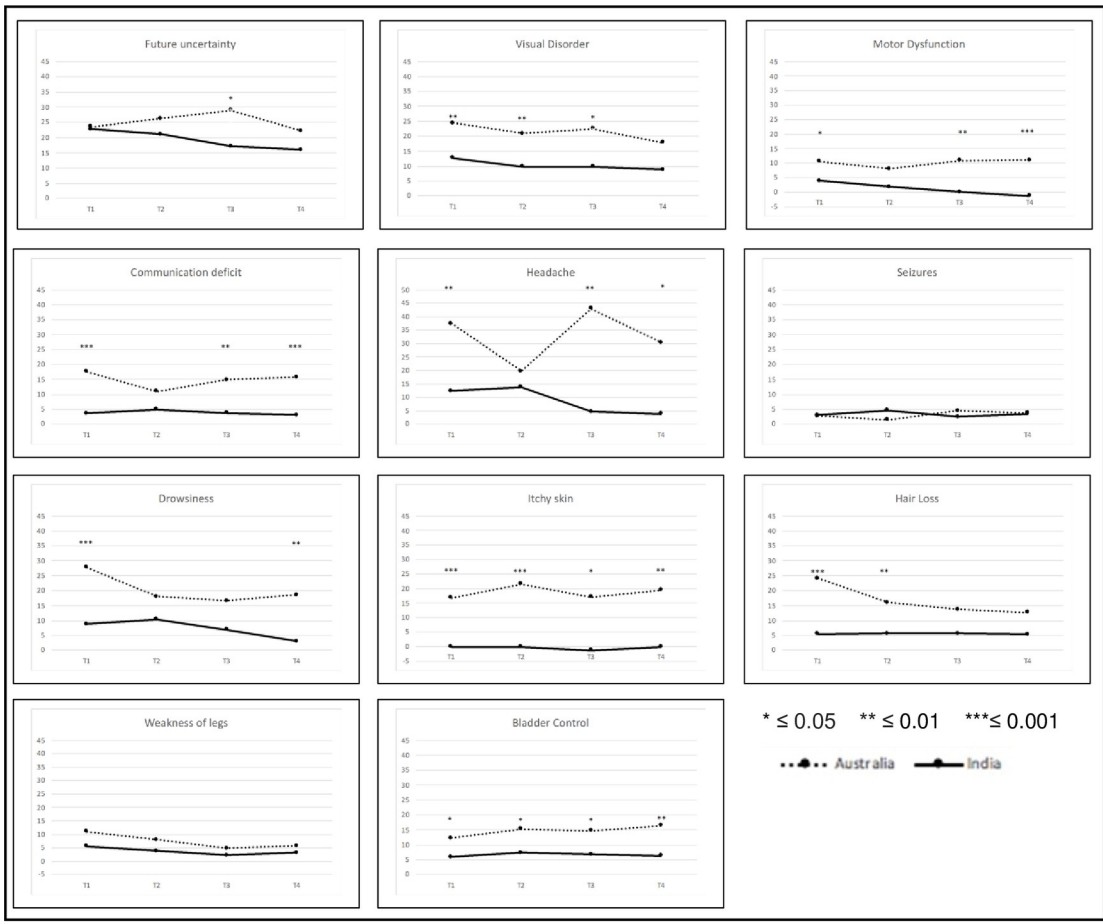

**Fig 3. BN20 scores.**

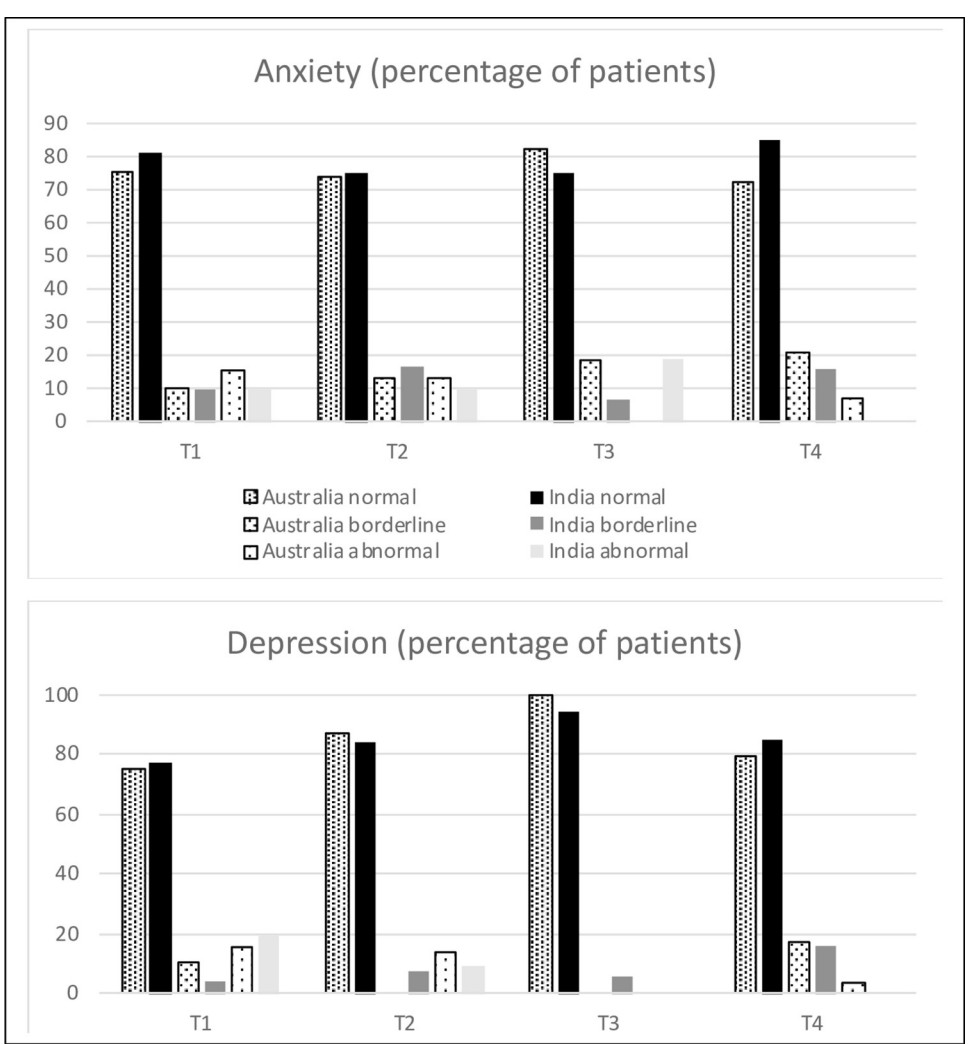

**Fig 4. Hospital and Anxiety Depression Scale (HADS)–frequencies of symptoms.**

legs were not statistically significantly different at all time points. Of note, due to the low occurrence of some symptoms, even a small difference may be statistically significant.

**HADS.** In general, anxiety and depression were uncommon over the 24 month postoperative period with >70% of patients showing no symptoms of anxiety and >75% of patients showing no symptoms of depression in both groups (Fig 4 and S4 Table). The mixed model analysis identified no statistically or clinically meaningful differences (+/- 1.6) between the countries for anxiety and depression ratings (S5 Table).

## Discussion

This is the first reported international comparison of postoperative HRQoL in brain tumour patients. Our analysis of HRQoL, symptom burden, anxiety and depression in Australian and Indian samples showed interesting differences between the samples and, at times, perhaps surprisingly similar results, despite expected cultural and socio-economic differences. As expected, HRQoL in meningioma patients was higher and symptom burden lower compared to reports of other brain tumours with more devastating survival outcomes [32, 33]. However,

in comparison to a normative European population, QLQ-C30 scores were still reduced for both the Australian and Indian population [31]. Differences between the study samples can roughly be divided into Australian patients reporting better physical function in the early months postoperative, with Indian patients slowly improving over time, whereas Indian patients report better global HRQoL and emotional functioning at 12–24 months postoperative, with Australian patients reporting a persistent deficit. Insomnia and fatigue were greater issues for patients in Australia and gastro-intestinal symptoms (nausea/vomiting, appetite loss, constipation, diarrhea) were more reported by patients in India over the course of the 2-year follow-up. Future uncertainty and symptoms common for brain cancer patients were consistently reported more commonly in the Australian group. No differences for depression and anxiety were identified.

## QLQ-C30 function scales

Global HRQoL, physical and cognitive function showed relatively stable ratings in the Australian population which is in keeping with the prolonged stable deficits reported in our previous work [25] but the marked improvement in the Indian population challenges this is as a generalizable conclusion and supports the need for further culturally diverse studies. Emotional function appears similar in both samples with the exception of T4, 12–24 months postoperative, with Australian patients deteriorating and Indian patients strongly improving.

Some of these differences may be explained by the different support systems that patients experience in their respective national, community and family frameworks. Regular follow-up care by outpatient clinics for Australian patients is well established in the first months after surgery but later on the patient relies on personal support systems, which are less present in the Australian society. In the Kerala region, regular follow-up care by health professionals is coupled with the benefits from long-term care provided by family members and their "therapeutic participation" (p. S299) [14]. Higher ratings of social function and the great improvement in emotional function and perceived global HRQoL in the Indian sample support this assumption.

## QLQ-C30 symptom scales and BN20

Our results demonstrated that symptom burden in both samples remained at low levels or improved over time, which is a common observation in other studies of meningioma patients [8, 34]. However, the consistently higher symptom burden in the Australian sample stands out. Our Indian sample reported particularly low symptoms on the BN20, even when compared to another Indian sample of patients with benign brain tumours or low-grade gliomas also assessed with the BN20 [35].

A possible explanation is the culturally intrinsic reporting of symptoms. It has been shown that recognition, intensity, interpretation and reporting of symptoms depends on cultural upbringing and learned behaviour within societies [36, 37]. While symptoms can be seen as an indicator for illness, individuals often choose to ignore symptoms or interpret differently what requires (medical) attention [36]. The culturally unique ways of expressing symptoms may explain some differences of how patients in Australia and India reported their symptom burden in our study. It could also be surmised that the likelihood of ongoing financial government support and free medical care in response to reported symptoms could play a role in higher symptom reporting by Australian patients. In contrast, patients in India with out-of-pocket expenses for ongoing support arrangements, may suffer financially from reporting their symptoms, that may subsequently be investigated.

## HADS

In line with a Norwegian study of 54 meningioma patients, anxiety and depression did not pose a particular burden on meningioma patients in our study [38]. The percentage of patients in our samples suffering from anxiety and depression are similar to a normative population, which shows values between 5–24% for anxiety and 5–9% for depression [39, 40].

Results were more similar in both countries than expected based on the differences reflected in the Human Development Index, which puts Australia at rank 6 and India at rank 129 [41]. Our findings may reflect the context of the Indian sample; comprising patients from Kerala with high literacy and health consciousness [42] and willingness and ability to spend on healthcare, even when not covered by insurance. At the same time, the Australian sample comprised patients from suburbs with relative socioeconomic disadvantage [43]. Nevertheless, the SES was significantly different between the groups. An Indian study of patients with benign tumours and low-grade gliomas identified that illiterate patients had significantly lower HRQoL scores [35]. Our Indian sample did not include any illiterate patients, another reason that may explain less than expected differences between countries.

## Study limitations

This collaborative project enabled us to directly compare HRQoL and symptom burden in Australian and Indian patients who underwent meningioma surgery in very different settings, which is the first of its kind.

Data for the Indian and Australian samples were collected in separate HRQoL studies and data collection for this joint study was not planned a priori, which is why patient numbers and data collection times differ. While we were able to control for potential confounding factors such as SES, employment and relationship status, we do not have information on the level of education, income, living conditions and social capital (that is, an individual's resources derived from social connections [44]). Ideally, a comparison of two more contrasting regions in regard to socioeconomic status could have identified more differences, however, neurosurgery would most likely not be available to brain tumour patients in less developed regions.

## Conclusion

Our international comparison of HRQoL and symptom burden following meningioma resection showed differences between Australian and Indian patients, but also some remarkable similarities. The low number of CMD demonstrates that overall meningioma patients in both samples may have similar experiences in terms of recovery, well-being and reductions in their HRQoL and functioning, although in general, the Indian sample perceived less long-term impairments. In particular mental health appears to be minimally affected by the diagnosis of a meningioma.

Routine and repeated assessments of HRQoL in meningioma patients is recommended to provide targeted follow-up care to improve reductions in HRQoL following meningioma treatment. Our data suggest that perceptions of HRQoL vary in different social settings, which highlights the importance of individual assessments to tailor support needs and maximise return to normal function and social participation.

The country-specific follow-up care and support systems (institutionalized/individual-dependent versus professional follow-up plus family-care/collectivist) in which patients recover from meningioma resection are not reflected in vastly different ratings of HRQoL outcomes and symptom burden. However, a qualitative study could identify the specific advantages and preferences of patients for their care and support systems. Future research should

investigate the association of long-term support from family members or other informal carers as a potentially important support system for meningioma patients.

## Supporting information

**S1 Checklist. Cultural inclusivity questionnaire.**
(DOCX)

**S1 Table. QLQ-C30 functional scales.**
(DOCX)

**S2 Table. QLQ-C30 symptom scales.**
(DOCX)

**S3 Table. BN20 scales.**
(DOCX)

**S4 Table. HADS percentages.**
(DOCX)

**S5 Table. HADS scales.**
(DOCX)

## Acknowledgments

Graham Hepworth from the Statistical Consulting Center of the University of Melbourne provided advice for statistical analysis. We also thank the medical students of the Department of Neurosurgery at the Royal Melbourne Hospital who collected QoL data for the Australian sample.

## Author Contributions

**Conceptualization:** Verena Schadewaldt, Sandhya Cherkil, Dilip Panikar, Katharine J. Drummond.

**Data curation:** Verena Schadewaldt, Sandhya Cherkil, Katharine J. Drummond.

**Formal analysis:** Verena Schadewaldt.

**Methodology:** Verena Schadewaldt, Katharine J. Drummond.

**Supervision:** Katharine J. Drummond.

**Visualization:** Verena Schadewaldt.

**Writing – original draft:** Verena Schadewaldt.

**Writing – review & editing:** Verena Schadewaldt, Sandhya Cherkil, Dilip Panikar, Katharine J. Drummond.

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
