## [Decision Letter · Decision Letter 0]

2 Aug 2022

PONE-D-22-10328Quality of life after resection of a meningioma – a cross-cultural comparison of Indian and Australian patientsPLOS ONE

Dear Dr. Schadewaldt,

Thank you for submitting your manuscript to PLOS ONE. After careful consideration, we feel that it has merit but does not fully meet PLOS ONE’s publication criteria as it currently stands. Therefore, we invite you to submit a revised version of the manuscript that addresses the points raised during the review process.

We look forward to receiving your revised manuscript.

Kind regards,

Ali Montazeri

Academic Editor

PLOS ONE

Journal Requirements:

Reviewers' comments:

Reviewer's Responses to Questions

**Comments to the Author**

1. Is the manuscript technically sound, and do the data support the conclusions?

Reviewer #1: Yes

Reviewer #2: Yes

2. Has the statistical analysis been performed appropriately and rigorously? 

Reviewer #1: Yes

Reviewer #2: Yes

3. Have the authors made all data underlying the findings in their manuscript fully available?

Reviewer #1: No

Reviewer #2: Yes

4. Is the manuscript presented in an intelligible fashion and written in standard English?

Reviewer #1: No

Reviewer #2: Yes

5. Review Comments to the Author

Reviewer #1: Thank you for the opportunity to review the manuscript. I reviewed this manuscript carefully with a great interest. I respectfully provided my comments below

1:The limitation of the study should be written shorter.

2:The study Conclusion should be written shorter.

3: The conclusion of the study should be written shorter

3:It is suggested that the results shown in the figure are given in the tables.

Reviewer #2: Review of: Quality of life after resection of a meningioma – a cross-cultural comparison of Indian and Australian patients

This is a well-written and interesting manuscript. However, I have some minor comments.

The introduction is discussion too long. Please summarize this part while maintain the focus on the research question.

The sentence" Additionally, accounting for international differences in health care systems, economic status and individual lifestyles that may affect HRQoL is generally not possible in systematic reviews" needs citation.

Method: Clearly define all outcomes, predictors, potential confounders, and effect modifiers (if any). Did the study have any missing data? How did you approach them?

Results

Add a table describing the mean and SD of all measures and related sub-scales according to time-point and groups of concern.

The discussion section is too long. Please revise.

Some references are too old, please replace with newer ones. The reference list should include references published over the last 5 years.

6. PLOS authors have the option to publish the peer review history of their article (what does this mean?). If published, this will include your full peer review and any attached files.

Reviewer #1: No

Reviewer #2: **Yes: **Marzieh Araban

---

## [Author Response · Author response to Decision Letter 0]

14 Aug 2022

Dear Reviewers and Editors,

We appreciate the reviewer feedback and understand the value of the time they took to review our manuscript. Their comments and suggestions have allowed us to improve our work. Please find our detailed responses to review comments in the attached file 'Responses to reviewers'.

We have prepared our article according to journal guidelines and updated file names and the reference list. We completed the inclusivity in global research questionnaire, which can be accessed as a supplementary file. Figure files will be provided in the requested format if the manuscript is accepted for publication.

Thank you for considering this manuscript for publication.

Kind regards,

Verena Schadewaldt

---

## [Editor Report · Decision Letter 1]

12 Sep 2022

Quality of life after resection of a meningioma – a cross-cultural comparison of Indian and Australian patients

PONE-D-22-10328R1

Dear Dr. Schadewaldt,

We’re pleased to inform you that your manuscript has been judged scientifically suitable for publication and will be formally accepted for publication once it meets all outstanding technical requirements.

Kind regards,

Ali Montazeri

Academic Editor

PLOS ONE
---

## [Editor Report · Acceptance letter]

16 Sep 2022

PONE-D-22-10328R1 

Quality of life after resection of a meningioma – a cross-cultural comparison of Indian and Australian patients 

Dear Dr. Schadewaldt:

I'm pleased to inform you that your manuscript has been deemed suitable for publication in PLOS ONE. Congratulations! Your manuscript is now with our production department. 

Kind regards, 

on behalf of

Professor Ali Montazeri 

Academic Editor

PLOS ONE